# A Novel Simplified System to Estimate Lower-Limb Joint Moments during Sit-to-Stand

**DOI:** 10.3390/s21020521

**Published:** 2021-01-13

**Authors:** Seoyoon Hwang, Seoyoung Choi, Yang-Soo Lee, Jonghyun Kim

**Affiliations:** 1School of Mechanical Engineering, Sungkyunkwan University, Suwon 16419, Korea; sy.hwang@skku.edu; 2Department of Robotics Engineering, DGIST (Daegu Gyeongbuk Institute of Science and Technology), Daegu 42988, Korea; sychoi@dgist.ac.kr; 3Department of Rehabilitation Medicine, School of Medicine, Kyungpook National University, Daegu 41944, Korea; leeyangsoo@knu.ac.kr

**Keywords:** asymmetric weight-bearing motion, inertial sensor, joint moment, kinematic model, segment angle, sit-to-stand

## Abstract

To provide effective diagnosis and rehabilitation, the evaluation of joint moments during sit-to-stand is essential. The conventional systems for the evaluation, which use motion capture cameras, are quite accurate. However, the systems are not widely used in clinics due to their high cost, inconvenience, and the fact they require lots of space. To solve these problems, some studies have attempted to use inertial sensors only, but they were still inconvenient and inaccurate with asymmetric weight-bearing. We propose a novel joint moment estimation system that can evaluate both symmetric and asymmetric sit-to-stands. To make a simplified system, the proposal is based on a kinematic model that estimates segment angles using a single inertial sensor attached to the shank and a force plate. The system was evaluated with 16 healthy people through symmetric and asymmetric weight-bearing sit-to-stand. The results showed that the proposed system (1) has good accuracy in estimating joint moments (root mean square error < 0.110 Nm/kg) with high correlation (correlation coefficient > 0.99) and (2) is clinically relevant due to its simplicity and applicability of asymmetric sit-to-stand.

## 1. Introduction

Sit-to-stand (STS), the act of rising from a chair is regarded as an important movement to use in functional assessments [1,2] because it is a prerequisite of many activities in daily living [3,4,5,6]. With the elderly [7] and mobility-limited patients (e.g., patients with hemiparesis [8,9], hip arthroplasty [6,10,11], hip fracture [12]), their STS becomes slow and accompanies asymmetric weight-bearing (asymmetric STS) due to the reduction of unilateral muscle strength in the lower limbs [9,13]. Especially, asymmetric STS relates to a decline in functional performance [13] and a higher fall risk [14]. Hence, the evaluation of STS (including asymmetric STS) is important for diagnosis and effective intervention.

During STS evaluations, the profile of joint moment (JM) or support moment (the summation of the ankle, knee, and hip JMs [15]), and maximum JM have been widely used as the major outcomes in studies for understanding STS motion. In an analysis of asymmetric STS, motion asymmetry has been evaluated using the JM to understand impaired motion due to illness [16,17], and to monitor the training effect after rehabilitation therapy [18,19] and surgery [18]. Moreover, there are several indices that are calculated according to JM and/or maximum JM to analyze or evaluate patients. The symmetric index (SI) for hips or knees is obtained using maximum JM [10,16,19], and the maximum support moment [20,21] can be applied to asymmetric STS analysis. Note that the maximum support moment has also been used in a different form, called the muscular strength margin, to evaluate muscular strength differences by age difference [22].

As the gold standard for obtaining the JM in STS, an inverse dynamic (bottom-up) method using both a motion capture system (Mocap) and force plates has been widely used in various laboratory setups [23]. However, these instruments together are inappropriate for a clinical environment because of their high cost, inconvenience to setup, and the requirement for a large amount of space [24]. As an alternative way to obtain JM, the motion analysis system based on small and cost-effective sensors (i.e., inertial sensor [23,25] and depth sensor [26]) was suggested. Note that some studies attempted to use the Wii balance board [13,27] and an instrumented chair with load cells [28] as a substitute for the force plate, but they were not to obtain JM.

Those motion analysis systems are mostly based on a top-down inverse dynamic method, this means they calculate JM from the top segment (head-arm-torso) to the bottom segment (foot) without Mocap and the force plate. However, owing to the lack of the force plate, these methods (1) show poor accuracy [25], (2) cannot obtain the JM prior to the subject no longer being in contact with the seat (seat-off) [23], and (3) are not able to cover asymmetric STS due to the assumption of symmetry movement and loading occurring during STS [26]. It has been reported that the maximum JMs in the knee and hip often appear before seat-off [29], thus the calculation of JM prior to seat-off is important. Several studies have attempted to overcome these drawbacks by re-defining the head-arm-torso as multiple-segments [23] and by predicting the JM before seat-off [25]. However, these methods still reported limited accuracy and relied on the assumption of symmetry [23,25,26]. This shows the fundamental limitation of the top-down method to obtain the JM and implies the need for a simplified (cost-effective and convenient) bottom-up method, which can cover weight-bearing asymmetry in STS, that could be more relevant to clinical setup.

For clinically relevant analysis of STS, this paper proposed a novel JM (and maximum JM) estimation method with a simplified system for slow STS that includes asymmetric STS. Even though the proposed system was based on the bottom-up method, we were able to estimate JM without Mocap. For that, we used (1) two inertial sensors attached to both shanks along with force plates to cope with various weight-bearing and (2) a kinematic model of STS based on environmental parameters for a cost-effective bottom-up method. Moreover, we suggested (3) a restricted but effective range for estimating JM by considering where the maximum JM should appear during STS. Through symmetric and asymmetric STS experiments with sixteen healthy subjects, the proposed system shows a high estimation accuracy for JM, support moment, and SI.

## 2. Methods

### 2.1. Kinematic Model of Sit-to-Stand

Figure 1a describes a user’s sitting when using the proposed JM estimation system. In order to obtain the user’s shank angle (*θ_shank_*) and thigh angle (*θ_thigh_*) without Mocap, we developed a kinematic model of STS described by environmental parameters, which contained the distances between the joint centers (dots in Figure 1a) and the landmarks (stars in Figure 1a) which were determined by the apparatus of the proposed system, such as the force plate and chair [30]. The kinematic model was based on a two-link rigid body that consisted of shank and thigh segments (Figure 1a).

For a simplified kinematic model, we assumed that the links move in the sagittal plane (YZ plane in global frame) during STS, and the feet are fixed to the ground [31]. It should be noted that sagittal plane motion has been regarded as the core motion of STS, even for asymmetric STS [3,32,33,34]. Due to this assumption, the ankle joint position **p_ankle_** can always be represented by the environmental parameters in the YZ plane as {0, *d*_1_, *d*_2_}. Here, *d*_1_ is the measured *Y*-axis distance between the ankle joint center and the landmark (on the edge of the force plate), and *d*_2_ = 0.039 *H*, where *H* denotes height, according to anthropometric data [15].

#### 2.1.1. Segment Angle Calculation While Sitting

While sitting, the inverse kinematics of the STS model was used to obtain *θ_shank_* and *θ_thigh_*. To solve the inverse kinematics, hip joint position during sitting was calculated as follows:(1)$$p_{hip,sit}=\left\{0,d_{3}-d_{4},d_{5}+d_{6}\right\}$$
where *d*_3_ denotes the measured length between the chair landmark and force plate landmark; *d*_5_ the measured vertical distance between the chair and the force plate; *d*_6_ = 0.05 *H* by anthropometric data [15]; and *d*_4_ was estimated by using an experimental model constituting the gender (*X*) and the leg length (*L*) [35], as below:(2)$$d_{4}=c_{1}X+c_{2}L+c_{3}.$$
Here, *c*_1_, *c_2,_* and *c*_3_ are constants and *L* is measured between the anterior superior iliac spine and medial malleolus [35].

From the inverse kinematics, *θ_shank_* and *θ_thigh_* at sitting can be calculated as follows (Figure 2a):(3)$$\theta_{shank,sit}=\pi -\tan^{-1}\left(\frac{p_{rel,1}\cdot \hat{Z}}{p_{rel,1}\cdot \hat{Y}}\right)-\cos^{-1}\left(\frac{l_{thigh}^{2}-l_{shank}^{2}-{\left\|p_{rel,1}\right\|}^{2}}{-2l_{thigh}\left\|p_{rel,1}\right\|}\right),$$
(4)$$\theta_{thigh,sit}=\pi -\left\{\cos^{-1}\left(\frac{{\left\|p_{\mathrm{rel},1}\right\|}^{2}-l_{thigh}^{2}-l_{shank}^{2}}{-2l_{thigh}l_{shank}}\right)-\theta_{shank,sit}\right\},$$
with **p_rel,1_** = **p_hip,sit_** − **p_ankle_**. *l_thigh_* and *l_shank_* denote the length of thigh and shank segments, respectively; and $\hat{Y}$ and $\hat{Z}$ denote Y and *Z*-axis unit vectors, respectively. Here, *l_thigh_* was measured between lateral epicondyle and greater trochanter, and *l_shank_* between the center of the knee and ankle joint in the coronal plane.

#### 2.1.2. Shank Angle Calculating after Sitting

After the subject begins to stand up, it is not possible to use the inverse kinematic approach above to obtain *θ_shank_* and *θ_thigh_* because the body segments are not in contact with the chair. In this study, from *θ_shank,sit_* in (3), *θ_shank_* was calculated by integrating shank angular velocity along the sagittal plane measured by an inertial sensor attached to the middle of the shank [36,37] (Figure 2). Here, the integration was started at the beginning of the STS motion, which was detected by using a z-directional ground reaction force (GRF) [38]. It should be noted that the sensor was attached only to the shank rather than the thigh or to both, because (1) *θ_shank_* affects both JMs in contrast to *θ_thigh_* and (2) our target was to minimize the number of sensors used for simplicity. Figure 2b shows a representative result of *θ_shank_* calculation.

#### 2.1.3. Thigh Angle Calculation After Sitting

We applied a regression method based on two connected cubic polynomials, one from sitting to seat-off and the other from seat-off to stand-up, to calculate *θ_thigh_*. The connection at seat-off between the polynomials was to improve *θ_thigh_* accuracy around seat-off, which is a key milestone in STS; the maximum JM occurred near seat-off [29].

In order to specify the polynomials, {*θ_thigh_*, $\dot{\theta }$*_thigh_*, *t*} at sitting, seat-off, and standing are required, respectively [39]. For sitting and standing, we used *θ_thigh,sit_* obtained from Equation (4), *θ_thigh,stand_* = 85.2° [40], and $\dot{\theta }$*_thigh_**_,sit_* =$\dot{\;\theta }$*_thigh_**_,stand_* = 0. Since force sensing resistors (FSR) to detect seat-off were attached to the chair (Figure 2a), the thigh angle at seat-off (*θ_thigh,off_*) could be obtained from an equation for the distance between the FSR landmark (**p_FSR_** = {0, *d*_7_, *d*_8_}) and the thigh segment line that passes through the knee joint position **p_knee,off_** [41] if we knew the distance *d*_9_ between the thigh segment line and the FSR landmark, as follow:(5)$$\theta_{thigh,off}=\pi -\tan^{-1}\left(\frac{(p_{\mathrm{rel},2}\cdot \hat{Z})(p_{\mathrm{rel},2}\cdot \hat{Y})+d_{9}\sqrt{{\left\|p_{\mathrm{rel},2}\right\|}^{2}-d_{9}{}^{2}}}{p_{\mathrm{rel},2}\cdot \hat{Y}}\right),$$
with **p_rel,2_** = **p_FSR_** − **p_knee,off_**. Note that **p_knee,off_** can be obtained by using **p_ankle_**, *l_shank_*, and *θ_shank,off_*. Here, based on the idea that the distance *d*_9_ in Equation (6) would be closely related to body thickness, *d*_9_ was obtained from the following experimental model:(6)$$d_{9}=c_{4}H+c_{5}W+c_{6},$$
where *W* denotes the weight; and *c*_4_, *c_5,_* and *c*_6_ the constants.

We calculated $\dot{\theta }$*_thigh,_**_off_* as the average of angular velocities from sitting to seat-off and from seat-off to stand. Time *t* at sitting, seat-off, and standing were detected using the force plate and FSR as follows. Sitting was detected when the z-directional GRF was larger than 103% of the initial sitting state; standing was detected when the total z-directional GRF (sum of left and right GRF) was in the 99~101% range for body weight [38]; seat-off was detected when all FSR data reached beyond the range of the standard deviation of the unloading state.

A typical result of the polynomial regression above is illustrated in Figure 2. Despite the numerous assumptions made, one can see that *θ_thigh_* after sitting could be obtained without Mocap and/or additional sensors (Figure 2).

### 2.2. Inverse Dynamics

To calculate JM using the bottom-up method, we used the inverse dynamics with the assumption that linear/angular accelerations were negligible [42,43] because the target task of this paper was slow STS with the elderly and mobility-limited patients. Note that all calculated JMs were normalized to the weight.

Although JM can be calculated within the whole range of STS, we suggested a restriction of the JM estimation range. This was because segment angle estimation errors could be enlarged after seat-off, which affected the accuracy of the JM estimation. As with *θ_shank_*, a greater drift effect in integrating angular velocity happened with a larger JM estimation range. Moreover, the estimation error for *θ_thigh_* tended to be significant after seat-off because a less-constrained motion after seat-off makes the regression-based estimation using cubic polynomials more inaccurate.

Despite the restriction of the range, we recommend including the location where maximum JM occurs for determining effective range because the maximum JM has been widely used to analyze asymmetric motion, such as SI [10,16,19]. For the ankles, maximum JM occurs near the end of the STS [29], thus the range for ankle JM can be regarded as the entire STS. On the other hand, the maximum JM for knee and hip joints usually occurs around seat-off [29] or sometimes after seat-off with hemiparesis patients [5]. Therefore, the effective JM estimation range for the knee and the hip can be represented to include the location of maximum JM, as follow:(7)$$T=\left[t_{start},\frac{\alpha }{100}(t_{end}-t_{start})+t_{off}\right],$$
where α denotes the constant percentage to determine the range after seat-off; *t_start_*, *t_end_*, and *t_off_* denote the times of start, end, and seat-off in STS start, respectively.

## 3. Experiments

### 3.1. Experimental Design

An experiment was conducted to evaluate the estimation accuracy of the JM using the proposed system. To implement the proposed system, two inertial sensors (Shimmer 3, Shimmer, Dublin, Ireland), two force plates (OR6-7, AMTI, Watertown, MA, USA), a height-adjustable chair, and four FSR sensors (Trigno FSR, Delsys, Natick, MA, USA) attached to the edge of the chair were used in the experiments (Figure 1b). The data from the inertial sensors were acquired using LabVIEW (National Instruments, Austin, TX, USA) with a 204.8 Hz sampling frequency. The data from the force plate and the FSR sensor were collected using Nexus (Vicon, Oxford, UK) at 1000 Hz. We used a Mocap as the gold standard to compare our results, the system used in the experiments consists of eight infrared cameras (Bonita, Vicon, Oxford, UK) and two identical force plates. Marker data for Mocap were acquired by using Nexus at a sampling frequency of 250 Hz. A DAQ board (USB-6211, National Instruments, Austin, TX, USA) was used to send a trigger to Mocap for data synchronization.

Sixteen healthy subjects (8 men, 8 women) participated in the experiment. Their profiles are summarized in Table 1. The experiment was conducted with the approval of the institutional review board (DGIST-170414-HR-007-01).

### 3.2. Protocols

Before the experiment, an inertial sensor was attached to each side of the shank by an elastic band. For Mocap, 34 reflective markers were placed on the subject’s body by following the guidelines of the Coda model for the pelvic region and the Visual 3 D model for the lower limbs [44]. To ensure STS began from a regular sitting posture among subjects, the height of the chair and foot position was adjusted for each subject. The height of the chair was changed to about 80% of the subject’s knee height [45] because weight-bearing is affected by the ratio of chair height and knee height [46], and the foot position was adjusted for each subject to make the subject’s initial *θ_shank_* to 75° using a digital goniometer. After that, we measured each segment length, *L* in Equation (2), *d*_1_ in Figure 2a, and the positions of the landmarks on the chair in Figure 1a using a measuring tape.

The experiments were conducted with the subjects performing STS with different weight-bearings. For the first task, we asked the subjects to perform symmetric (natural) STS. In the second task for asymmetric STS, we asked the subjects to perform STS with more weight-bearing on their dominant hand side as much as they could possibly manage. During the experiment, the subjects were instructed to have their back-touching the backrest of the chair during sitting and to fold their arms during STS. All tasks were replicated three times.

### 3.3. Data Analysis

Collected Mocap and force plate data were filtered using a low-pass bi-directional Butterworth filter with 6 Hz and 20 Hz cut-off frequencies, respectively. Based on the joint positions obtained by the Visual3D software (C-motion, Germantown, MD, USA) using the Mocap data, we calculated the segment angles as the kinematic golden standard to compare our system’s results. The JMs were computed from the Mocap and the force plate data through Visual3D. The FSR sensor data was post-processed using a moving average filter with a 10 window size, and the inertial sensor data was filtered using the low-pass Butterworth filter with a 10 Hz cut-off frequency. Note that the trigger signal data were not filtered. All collected data were interpolated to 204.8 Hz. The parameters of the experimental models’ Equations (2) and (6) used for the proposed system were determined through linear regressions (R^2^ = 0.82 for Equation (2) and 0.57 for Equation (6)) using 8 of 16 subjects’ *d*_4_ and *d*_9_ along with *X*, *L*, *H*, and *W*, as summarized in Table 2.

In this paper, α in (7) for the effective JM estimation range was chosen as 11(%) [47], based on the idea that the maximum JM appears before a decrease in horizontal momentum. To evaluate the accuracy of the proposed JM estimation system, the root mean square error (RMSE) and correlation coefficient (CC) were calculated within the JM estimation range. The results of the asymmetric STS task were separately analyzed according to the level of weight-bearing: less weighted side (non-dominant hand side) and more weighted side (dominant hand side). To eliminate the effect of the weight-bearing, the RMSE of the JM was also normalized by the RMS of the JM from the Mocap result. In addition, for evaluating the applicability of the proposed method to asymmetric STS, we conducted paired t-test analysis through SPSS (IBM, Armonk, NY, USA) to compare the accuracies between symmetric and asymmetric STS tasks.

For knee and hip joints, each SI was calculated as the ratio of the estimated maximum JM on the less weighted side to maximum JM on the more weighted side [16,19]. The accuracy of the SI was evaluated by comparing the SI with the SI calculated from the actual maximum JMs captured using Mocap. Moreover, the estimates for other major STS outcomes (i.e., maximum JMs and maximum support moment [16,17,18,19,20,22]) were evaluated by calculating the absolute error and the absolute percentage error for each outcome as below:(8)$$\frac{\left|M_{esti,i}-M_{con,i}\right|}{M_{con,i}}\cdot 100,$$
where *M_esti,i_* (*i* = ankle JM, knee JM, hip JM, support moment) denotes the maximum JM (or support moment) estimated by the proposed system in the JM estimation range and *M_con,i_* denotes the maximum JM (or support moment) computed using Mocap and force plates over the whole STS range. In addition, the timing differences between *M_esti,i_* and *M_con,i_* were calculated for the evaluation.

We calculated the RMSE and CC for the segment angles (*θ_shank_* and *θ_thigh_*) to evaluate the segment angle calculations of the proposed system. In addition, the absolute errors of *θ_shank,sit_*, *θ_shank,off_*, *θ_thigh,sit_*, and *θ_thigh,off_* were calculated to investigate the cause of JM estimation errors.

## 4. Results

### 4.1. Estimation of Joint Moments

Figure 3 shows a comparison of the mean lower-limb JMs and support moment for the proposed system with the results from conventional Mocap. One can see that the estimated JMs and support moment torque from the proposed system were highly correlated with those measured by the conventional system (Figure 4). The insignificant RMSEs (≤0.103 for JMs, ≤0.102 for support moment) and the very high CCs (≥0.990 for JMs, ≥0.997 for support moment), which are summarized in Table 3, quantitatively show our system offers affordable, accurate JM estimation. The RMSE of the ankle JM was the smallest compared to other JM estimates for the knee and hip.

A comparison of JM and support moment estimation accuracies between symmetric and asymmetric STS tasks are displayed in Figure 4. Despite the difficulty estimating in asymmetric STS, there was no statistical difference (*p* > 0.05) between the levels of accuracy, except for the knee JM (*p* = 0.01) on the less weighted (non-dominant hand) side only.

### 4.2. Estimation of Other Sit-to-Stand Outcomes

Estimation accuracy of other major STS outcomes, maximum JMs, and maximum support moment, are also summarized in Table 4. On average, the absolute (percentage) errors of the maximum JMs and maximum support moment were below 0.17 Nm/kg (15%) and 0.09 Nm/kg (10%) in all tasks, respectively (Table 4). The estimation of the maximum ankle JM was the most accurate, like JM estimation. As to the estimated timing of the outcomes, the timing differences were below 0.3 for the maximum JMs and 0.16 for the maximum support moment (Table 4). There was no significant difference in the estimate errors for symmetric STS, and the more/less weighted side of asymmetric STS.

Figure 5 shows scatter plots of the SIs comparing the proposed system with the gold standard Mocap results. Note that those SIs were calculated for hip and knee joints during the asymmetric STS tasks [16,18]. The errors between the two SIs were quite small (0.055 ± 0.056 for knee joint and 0.051 ± 0.046 for hip joint), and the relationship between the two SIs was well fit to the unity slope line (R^2^ = 0.825 for knee joint, R^2^ = 0.947 for hip joint), as shown in Figure 5.

### 4.3. Estimation of Segment Angles

Table 5 shows a comparison of the segment angles obtained from the proposed system and those from the conventional system. The RMSEs for *θ_shank_* and *θ_thigh_* were less than 4° and 6°, respectively, and the CCs for both segments were higher than 0.9 for all tasks. The mean absolute errors for *θ_thigh,sit_* and *θ_thigh,off_* were less than 5.5° while the errors for *θ_shank,sit_* and *θ_shank,off_* were less than 4.5°.

## 5. Discussion

We propose a novel system for JM estimation during STS, the system uses force plates and inertial sensors. This system does not require Mocap and uses a single inertial sensor while the previous state-of-the-art studies required three or four inertial sensors for each side of the body [23,25]. Thus, to our knowledge, the proposed system uses the simplest apparatus which is easily attainable. Moreover, compared with previous studies, the system can measure JM and JM-related indices during asymmetric STS that commonly appear in elderly and mobility-limited patients. Therefore, the proposed system has the potential to be widely used in a clinical setup.

Despite its simplicity, the proposed system shows a smaller JM error (<0.075 Nm/kg) than in existing studies (<0.13 Nm/kg) for symmetric STS [23]. Considering the difference of the range analyzed, the result, at least, implies the comparable JM estimation performance to the existing studies. Note that some ankle JM errors appeared due to the difference of ankle joint definition between the proposed system (fixed ankle model) and Mocap (floating pelvis model) [48]. Moreover, the estimated support moment of the proposed system was also quite accurate. The estimation error for the support moment was smaller than for all JM estimates. In addition, the system can accurately provide the following outcomes related to STS: maximum JM, maximum support moment, and SI. It is noteworthy that, to our knowledge, there is no existing study that estimates those outcomes in STS.

In contrast to previous studies, the proposed system can estimate JMs prior to seat-off. This is valuable in STS analysis, especially for the elderly and mobility-limited patients. Elderly and injured patients, due to their balance impairment and/or muscle weakness, stoop to move their center of mass from the chair to their feet before seat-off [9,14,49], and thus, their JM profiles before seat-off show differences from healthy people, as reported in a study on the STS characteristics of patients with Parkinson’s disease [50,51].

The proposed system can estimate JMs during both symmetric and asymmetric STS while existing studies based on a top-down method cannot cover asymmetric STS [23]. Hence, this study proposes the first attempt to propose a simplified method to estimate JMs in asymmetric STS. The JM estimating performance during asymmetric STS is comparable to that during symmetric STS. The JM error of our system for asymmetric STS (<0.103 Nm/kg) was smaller than the errors seen in existing studies for symmetric STS [23], in addition, most of the paired t-test results showed no difference in performance between symmetric and asymmetric STS (Figure 4). Those results show that our simplified approach based on the sagittal motion is an effective approximation for asymmetric STS, which would contain non-sagittal motion. It should be noted that the asymmetric STS tasks conducted in this study were designed to mimic abnormal STS which appears in the elderly and patients. The SI values in our asymmetric tasks (0.5 for knee and 0.48 for hip) were larger than those in another abnormal STS study that tested subjects’ total knee arthroplasty surgery (0.67~0.95 for knee and 0.95~0.99 for hip) [16,19].

The simplicity of the proposed system mainly comes from the segment angle estimations based on a kinematic model. The performance of segment angle calculations (RMSE 3.7° for the shank and 5.8° for thigh) is comparable to that of more complex systems (RMSE 2.0°~3.6° for the shank and 5.2°~6.5° for thigh) seen in existing STS studies [23,52], these studies included many more sensors with attaching issues. Furthermore, the performance was not degraded in asymmetric STS; the difference in the RMSE between symmetric STS and asymmetric STS was less than 0.3.

This study suggested the restricted JM estimation range for ensuring reliable JM estimation, including the maximum JM that usually occurs after seat-off. Even though we simply chose a constant α in (7), most peaks for hip/knee JMs and support moment measured by the gold standard setup (Mocap and force plates) in our experiment were located within the range chosen (about 92% of 576 STS data). This result shows that the effective JM estimation range (7) works and can be easily determined.

For the hip joint, the increase in JM estimation error after peak JM is not negligible (Figure 3). These errors are mainly caused by the degradation in *θ_thigh_* calculation performance after seat-off, which was supported by significant differences (*p* < 0.05) of the performance between before and after seat-off, as shown in Figure 6. Note that there were no significant differences for the knee and ankle joints (Figure 3). Despite this degradation caused by calculations based on polynomial regression, the estimated JMs and JM-related indices from the proposed system show adequate accuracy.

As mentioned before, a key advantage of the proposed JM estimation system is its simplicity making it a clinically relevant setup. Although the system still uses a force plate instead of a simplified force plate (i.e., Wii balance board [13,28] and K-FORCE [53]) as well as an inertial sensor, we believe that the proposed system is a well-optimized solution. The simplified force plate could not accurately measure Y-directional GRF and center of pressure, which are essential to estimate JM. The inertial sensor was attached to the shank to calculate *θ_shank_* during STS because knee JM estimation is sensitive to the error of *θ_shank_*. This sensitivity comes from the fact that the range of motion for *θ_shank_* (Figure 2b) is generally smaller than for *θ_thigh_* while the magnitude of the knee JM is similar to that of the hip JM (Figure 3).

This study still has room for improvement. The proposed system was not applied to the potential target group, elderly and patients with motor deficits, while its performance was evaluated with 16 healthy subjects who mimicked the target group by conducting asymmetric STS. The proposed system used two experimental models for kinematic modeling of STS (Table 2), while both models showed moderate to high correlation (R^2^ = 0.82 and 0.57) with healthy subjects. Hence, in future work, the proposed system needs to be validated with a larger population including the elderly and patients. Moreover, this study lacked a usability test of the proposed system while the protocol used in this study could be applied to the target population. Based on feedback from the usability tests with clinicians and target subjects, the proposed system could be made more suitable for use in a clinical setup, such as releasing the sitting constraint (i.e., increasing chair height [14,27,46]) used in this study.

## 6. Conclusions

In this paper, we developed a simplified JM estimation system for clinically relevant STS analysis, as a substitute for conventional Mocap-based setups. Thanks to the proposed kinematic model-based approach, the segment angles, which are required as input for the inverse dynamics to estimate JM, can be calculated from an inertial sensor without the need for Mocap, for cost-effective and convenient analysis for clinical use. In contrast to the existing attempts that also do not use Mocap, the proposed system can estimate JM (and JM-related indices) during asymmetric STS, this is essential for clinical use to analyze STS with the elderly and patients who have weight-bearing asymmetry during STS. By comparing our estimates with the results from the Mocap-based setup, it was shown that the proposed system has adequate JM estimation accuracy (RMSE: <0.110 Nm/kg; CC: >0.99), even under various asymmetric STS tasks. It should be noted that the level of accuracy was at least comparable to existing attempts. Hence, we believe that the proposed system could be a strong alternative to the conventional setup for wide clinical use. 

## Figures and Tables

**Figure 1 sensors-21-00521-f001:**
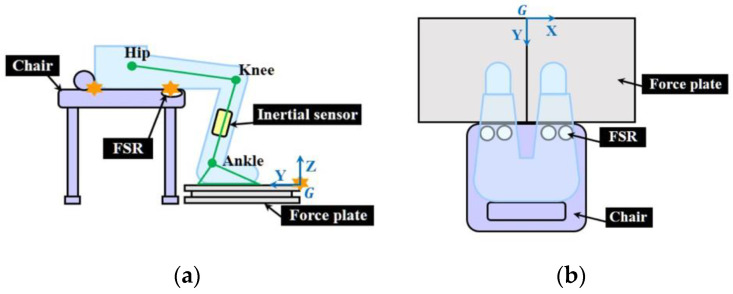
Schematic description of the proposed system in (**a**) YZ plane; (**b**) XY plane. Stars denote the landmarks for the system. FSR: force sensing resistor.

**Figure 2 sensors-21-00521-f002:**
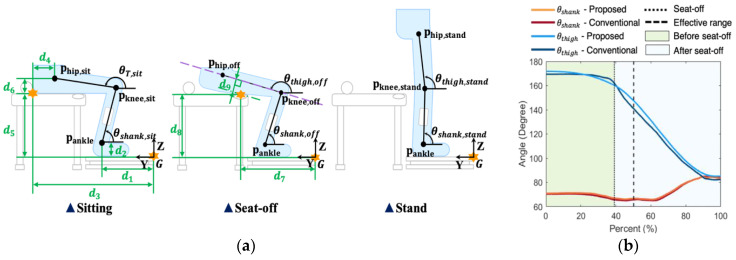
Kinematic model; (**a**) schematic diagram, (**b**) expected result of the kinematic model. Note that the environmental parameters (green) are defined by distances between landmarks (orange stars) and joint center and the purple dashed line at seat-off is the thigh segment line that passes through the knee joint position.

**Figure 3 sensors-21-00521-f003:**
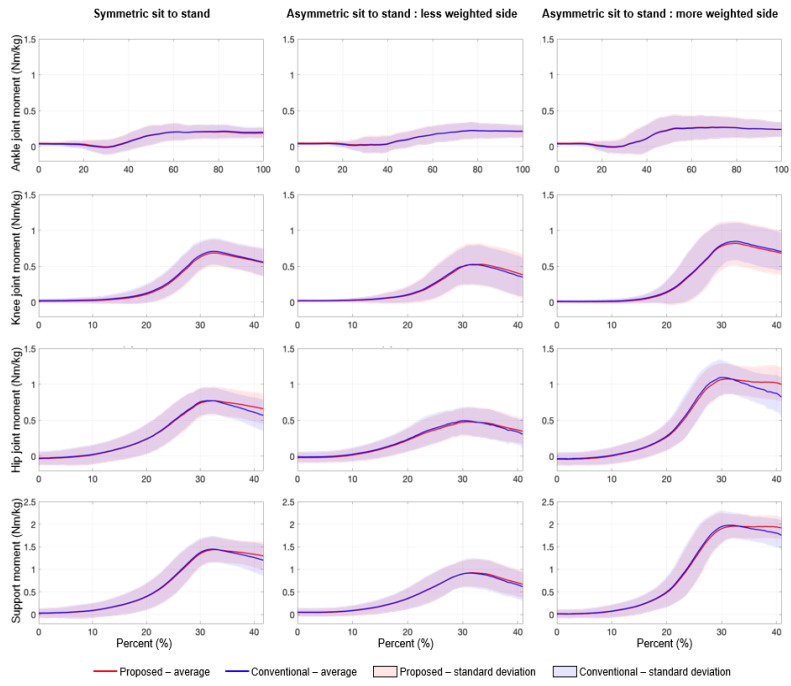
Comparison of joint moments and support moment estimated by the proposed system and measured by the conventional (Mocap-based) system. Note that knee and hip joint moments and support moment were obtained within the effective estimation range.

**Figure 4 sensors-21-00521-f004:**
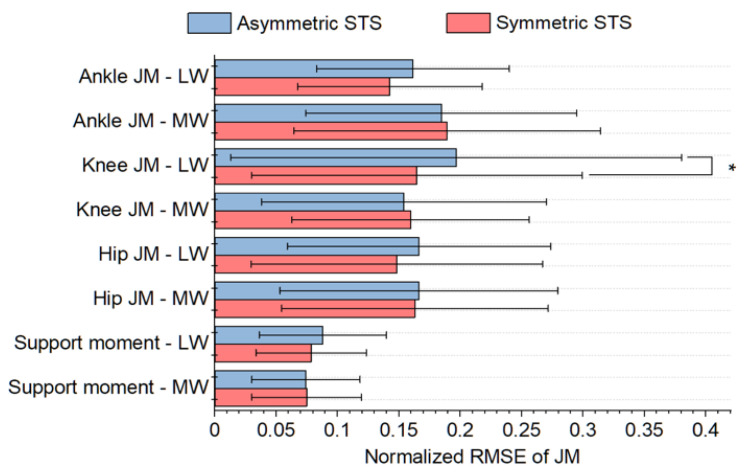
Comparison of RMSEs of normalized JMs and support moment between symmetric and asymmetric STS. Note that LW and MW denote the less weighted side and more weighted side, respectively. * indicate a significant difference (*p* < 0.05).

**Figure 5 sensors-21-00521-f005:**
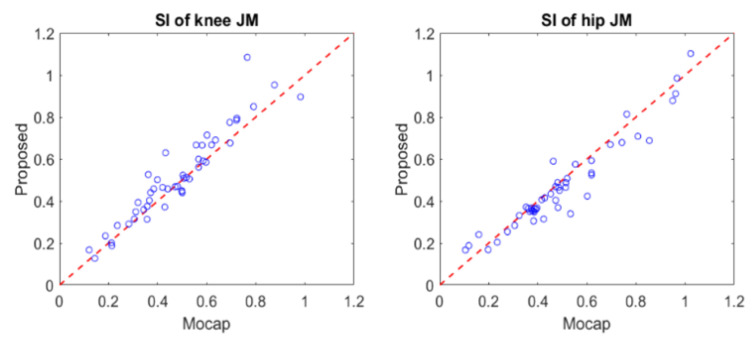
Comparison between the symmetric index (SI) estimated by the proposed method and measured by the motion capture system.

**Figure 6 sensors-21-00521-f006:**
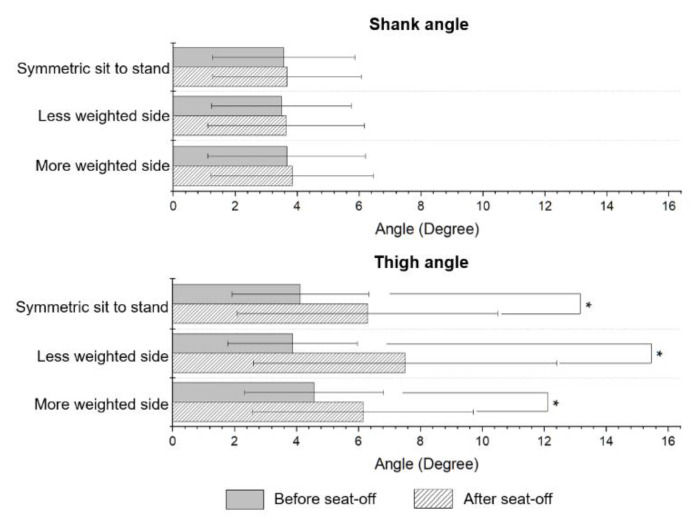
Comparison of RMSEs of segment angle between different STS ranges of shank angle and thigh angle. * indicates significant difference (*p* < 0.05).

**Table 1 sensors-21-00521-t001:** Summary of subject profiles.

		Mean (Standard Deviation)
Age (years)		27.563 (2.850)
Height, *H* (m)		1.676 (0.087)
Weight, *W* (kg)		63.588 (11.134)
Leg length, *L* (m)	Left	0.847 (0.062)
	Right	0.849 (0.061)
Shank length, *l_shank_* (m)	Left	0.379 (0.028)
	Right	0.380 (0.025)
Thigh length, *l_thigh_* (m)	Left	0.390 (0.020)
	Right	0.385 (0.037)

**Table 2 sensors-21-00521-t002:** Parameters of the experimental models’ equation.

Models	Constants
Equation (2)	*c*_1_ = 0.137, *c*_2_ = −0.026, *c*_3_ = 0.252
Equation (6)	*c*_4_ = 0.233, *c*_5_ = −0.00144, *c*_6_ = −0.175

**Table 3 sensors-21-00521-t003:** Comparison of joint moments and support moment obtained from the proposed system and conventional system with the motion capture system.

		RMSE (Nm/kg)	CC (95% CI)
Ankle joint moment	Sym.	0.026 ± 0.010	0.99 ± 0.01 (0.988–0.990)
	LW	0.024 ± 0.009	0.99 ± 0.01 (0.991–0.995)
	MW	0.028 ± 0.010	0.99 ± 0.01 (0.992–0.996)
Knee joint moment	Sym.	0.067 ± 0.048	1.00 ± 0.01 (0.997–0.999)
	LW	0.041 ± 0.031	0.99 ± 0.02 (0.990–0.998)
	MW	0.091 ± 0.058	1.00 ± 0.00 (0.999–0.999)
Hip joint moment	Sym.	0.068 ± 0.045	0.99 ± 0.01 (0.988–0.994)
	LW	0.044 ± 0.023	0.99 ± 0.01 (0.987–0.994)
	MW	0.105 ± 0.068	0.99 ± 0.01 (0.989–0.995)
Support moment	Sym.	0.066 ± 0.038	1.00 ± 0.00 (0.996–0.998)
	LW	0.045 ± 0.019	1.00 ± 0.00 (0.996–0.998)
	MW	0.090 ± 0.054	1.00 ± 0.00 (0.996–0.998)

CI denotes confidence interval. Sym. denotes symmetric sit-to-stand; LW and MW denote less weighted side and more weighted of asymmetric sit-to-stand, respectively. Values shown are mean ± standard deviation.

**Table 4 sensors-21-00521-t004:** Comparison of maximum joint moments and maximum support moment obtained from the proposed system and conventional system with the motion capture system.

		Absolute Error (Nm/kg)	Absolute Percentage Error (%)	Timing Difference (Second)
Maximum ankle joint moment	Sym.	0.024 ± 0.016	8.68 ± 5.84	0.285 ± 1.047
	LW	0.025 ± 0.018	10.79 ± 8.34	0.183 ± 0.636
	MW	0.025 ± 0.022	6.83 ± 6.02	0.047 ± 0.219
Maximum knee joint moment	Sym.	0.104 ± 0.079	13.13 ± 10.36	0.051 ± 0.175
	LW	0.064 ± 0.051	14.07 ± 13.78	0.077 ± 0.203
	MW	0.144 ± 0.094	14.21 ± 9.09	0.192 ± 0.435
Maximum hip joint moment	Sym.	0.096 ± 0.073	11.92 ± 9.02	0.122 ± 0.225
	LW	0.065 ± 0.048	12.76 ± 9.29	0.191 ± 0.314
	MW	0.164 ± 0.139	13.98 ± 12.28	0.172 ± 0.273
Maximum support moment	Sym.	0.086 ± 0.065	5.90 ± 5.08	0.130 ± 0.231
	LW	0.070 ± 0.046	7.52 ± 5.49	0.151 ± 0.308
	MW	0.136 ± 0.106	6.12 ± 4.68	0.131 ± 0.227

**Table 5 sensors-21-00521-t005:** Errors of segment angles in the proposed system.

		Shank Angle			Thigh Angle		
		Sym.	LW	MW	Sym.	LW	MW
Sitting	Absolute error (deg)	3.61 ± 2.42	2.93 ± 2.13	4.18 ± 2.62	4.90 ± 3.96	3.84 ± 2.30	5.32 ± 4.86
Seat-off	Absolute error (deg)	3.78 ± 2.72	3.66 ± 2.66	3.58 ± 2.69	3.69 ± 3.09	3.90 ± 2.91	4.67 ± 3.17
Effective range	RMSE (deg)	3.73 ± 2.41	3.58 ± 2.31	3.78 ± 2.48	5.19 ± 2.38	5.54 ± 2.21	5.29 ± 2.09
	CC	0.99 ± 0.02	0.99 ± 0.02	0.98 ± 0.03	0.95 ± 0.04	0.97 ± 0.03	0.94 ± 0.04

Sym. denotes symmetric sit-to-stand; LW and MW denote less weighted side and more weighted side of asymmetric sit-to-stand, respectively. Values shown are mean ± standard deviation. Note that absolute error was calculated for typical STS events, sitting and seat-off.

## Data Availability

The data presented in this study are available on request from the corresponding author. The data are not publicly available due to participant confidentiality.

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
