# Peer review of "A Novel Simplified System to Estimate Lower-Limb Joint Moments during Sit-to-Stand"

_sensors, 2021, doi:10.3390/s21020521_

Round 1

Reviewer 1 Report

The authors propose a new system for joint moment estimation of lower limbs during slow sit-to-stand movements. Their estimation method includes two inertial sensors and a kinematic model that can estimate both symmetric and asymmetric movements without the need of conventional cameras based motion capture systems. The system was validated by comparing their estimation results with those by using a motion capture system. The contribution is interesting with significant practical relevance.

I expected to understand why the height of the chair was changed for each subject. What could be the effect of using a fixed seat height on the symmetric index graphs? Could they differ significantly between the use of low and high seat heights?

It would be interesting to see how well the proposed approach works with elderly and patients with a real weight bearing asymmetry. Clearly, incorporating this additional analysis work would present a challenge, particularly since the paper is already quite comprehensive. Perhaps, the authors could offer a short discussion regarding these clinical issues.

Reviewer 2 Report

First of all, I would like to congratulate the authors for the article.

It is an interesting and clinically relevant topic.

I believe that the analysis performed is sufficient to start new research with elderly subjects or with different pathologies.

I only suggest presenting the clinical relevance more effectively during the introduction of the study, as well as at the conclusion.

Reviewer 3 Report

General comments:

The article could be interesting for the scientific literature, but it needs more hard working on it to become publishable.

Abstract:

The focus should be on the purpose of the study, the method used, the results obtained, the explanation in words, the discussion of the conclusions. In particular, it is necessary to explain the conventional equipment.

Introduction:

Line 72-74, I don’t understand the aim of your study. Is it to identify the accuracy of the proposed system by comparing the muscle strength and asymmetric weight load of the proposed system and the existing equipment?

Experiments

Line 178, Explain how to calculate the sample size of 16 subjects.

Line 192, A detailed explanation of the procedure for performing asymmetric STS is required.

Reviewer 4 Report

Please see the attached report.

Round 2

Reviewer 4 Report

I would like to thank the authors for the consideration given to the comments provided. The manuscript appears great.